# Canopy Architectural Characteristics of Ten New Olive (*Olea europaea* L.) Genotypes and Their Potential for Cultivation in Super-High-Density Orchards

**DOI:** 10.3390/plants13101399

**Published:** 2024-05-17

**Authors:** Marina Bufacchi, Franco Famiani, Valentina Passeri, Andrea Domesi, Adolfo Rosati, Andrea Paoletti

**Affiliations:** 1Istituto per i Sistemi Agricoli e Forestali del Mediterraneo, Consiglio Nazionale delle Ricerche, Via della Madonna Alta 128, 06128 Perugia, PG, Italy; marina.bufacchi@cnr.it (M.B.); valentina.passeri@cnr.it (V.P.); andrea.domesi@cnr.it (A.D.); 2Dipartimento di Scienze Agrarie, Alimentari e Ambientali, Università degli Studi di Perugia, Borgo XX Giugno 74, 06121 Perugia, PG, Italy; 3Centro di Ricerca Olivicoltura, Frutticoltura e Agrumicoltura, Consiglio per la Ricerca in Agricoltura e l’Analisi dell’Economia Agraria, Via Nursina 2, 06049 Spoleto, PG, Italy; adolfo.rosati@crea.gov.it (A.R.); andrea.paoletti1@gmail.com (A.P.)

**Keywords:** tree architecture, branching frequency, branching density, branch insertion angle, diameter ratio, olive orchard, cross-breeding

## Abstract

In recent years, there has been growing interest in olive genotypes (*Olea europaea* L.) suitable for super-high-density (SHD > 1200 trees/hectare) orchards. To date, only a few cultivars are considered fitting for such cultivation system. In this study, the first results on the architectural characteristics of the canopy of ten new olive genotypes are presented. Their suitability for SHD orchards was evaluated and compared with the cultivar ‘Arbequina’, which is considered suitable for SHD olive orchards and, for this reason, was used as the control. Several canopy measurements were taken, and some architectural parameters, such as branching frequency, branching density, and branch diameter/stem diameter ratio were calculated. The branching frequency value was greater than 0.20 in ‘Arbequina’ and in only four of the genotypes. The branching density in five genotypes was similar to ‘Arbequina’. ‘Arbequina’ had the lowest value for the branch diameter/stem diameter ratio, and only three genotypes had similar values. These initial results showed that only one genotype has all canopy architectural characteristics comparable to those of the cv. ‘Arbequina’. Further studies are needed to evaluate the production traits of these new genotypes and complete their characterization.

## 1. Introduction

The cultivated olive (*Olea europaea* subsp. *europaea*, var. *europaea*) is the second most worldwide oil fruit tree crop species. Extra virgin olive oil, EVOO, is one of the most valuable foods due to its high content of healthy fatty acids and minor constituents [1,2,3]. The genotype plays an important role in influencing the bioactive compound profile of olive fruits [4,5], and breeding programs can provide beneficial variability for EVOO quality, but also for agronomical characters useful for olive cultivation [6,7,8,9]. Therefore, it would be very useful to use breeding programs to find genotypes that combine high oil quality with agronomic characteristics that make them suitable for the new planting models that are being increasingly applied in olive growing.

In the last years, there has been increased interest in super-high-density (SHD) cultivation systems that have some important advantages, such as rapid achievement of full production, efficient mechanization of olive harvesting, and partial/full mechanization of pruning [9,10,11,12]. SHD orchards are characterized by close planting distances (the most used is 4 m × 1.5 m), thus requiring the use of low vigor and compact varieties [11,12,13,14]. In this regard, the study of tree architecture, which describes the habit of growth of trees, is crucial to understand the suitability of a genotype for the cultivation in high-density systems [15]. Indeed, the few studies carried out on the architectural features of the olive tree highlight that there is a great variability among genotypes [16]. Rosati et al. [13] provided a detailed description of the structural/architectural characteristics of trees which make them suitable for cultivation in SHD orchards and indicated the parameters that should be considered when evaluating the tree architecture. In particular, the proposed parameters include diameter and node number of the main stem (central leader) and number, as well as diameter and angle of insertion, of the main branches inserted in the central leader. These parameters allow us to characterize the growth habit of the genotype. In particular, the branching density (i.e., the number of lateral ramifications per unit length of the central axis) is especially important because it strongly affects the compactness of the canopy [17]: a high branching density, coupled with small shoot diameter, gives a higher compactness than low branching with thicker shoots. In this regard, it is also important to consider that branching seems also to be influenced by the propagation method utilized: micro-propagated olive trees showed a higher number of primary branches on the central axis than trees obtained by cutting [18]. Therefore, high branching and small diameters of the vegetation seem like important architectural characteristics that make cultivars suitable for SHD orchards.

At the time, there are very few cultivars that meet the requirements of the SHD system [12]. Therefore, it would be very useful to have more genotypes suitable for this kind of orchard. They could be identified and selected by investigating/evaluating the architectural characteristics of the available olive germplasm, also including minor varieties, and/or new genotypes produced by breeding. 

The aim of the present work was to evaluate the architectural characteristics of some new genotypes obtained from a breeding program in order to assess their potential for cultivation in SHD orchards.

## 2. Results

The average length of the main branches was the shortest (around 36 cm) in ‘I-79 free pollinated’ (G) (identification letters of the different genotypes are reported in Table 1), which was similar to ‘Arbequina’, while it was the longest (about 64 cm) in ‘Fs-17 × Bouteillan’ (A) (Figure 1). All other genotypes had values ranging from 40 to 50 cm in length.

The length of the internode evaluated on the same main branch varied from about 1.4 cm in ‘Bouteillan × Nostrale di Rigali’ (D) to just over 2 cm in the genotypes ‘Nociara free pollinated 1’ (H) and ‘Nucalia 1 × Don Carlo’ (L) (Figure 2). 

The angle of insertion of the main branches on the central leader was low (i.e., more vertical) for the genotypes ‘I-79 free pollinated’ (G), ‘Nociara free pollinated 1’ (H), and ‘Nociara free pollinated 2’ (I), with average values just under 50 degrees. This was lower than that of the cultivar ‘Arbequina’, while it was highest (i.e., more horizontal) for the genotype ‘I-77 × Kalamata’ (F), which showed values around 70 degrees (Figure 3). 

The branching frequency value was greater than 0.20 in ‘Arbequina’ and in the genotypes ‘Nucalia 1 × Don Carlo’ (L), ‘Nociara free pollinated 2’ (I), ‘I-79 free pollinated’ (G), and ‘I-77 self-pollinated’ (B). The genotypes ‘Fs-17 × Bouteillan’ (A), ‘Nucalia 1 × Nostrale di Rigali’ (C), and ‘Bouteillan × Nostrale di Rigali’ (D) showed a branching frequency value significatively lower than Arbequina and of the three genotypes ‘Nucalia 1 × Don Carlo’ (L), ‘Nociara free pollinated 2’ (I), and ‘I-79 free pollinated’ (G) (Figure 4). 

The order of genotypes for branching density was slightly different. In fact, ‘Nociara free pollinated 2’ (I) dropped compared to the others, while ‘Fs-17 × Cipro’ (E) gained (Figure 5). The genotypes ‘I-77 self-pollinated’ (B), ‘Fs-17 × Cipro’ (E), ‘Nucalia 1 × Don Carlo’ (L), ‘I-77 × Kalamata’ (F), and ‘I-79 free pollinated’ (G) had values statistically similar to that of ‘Arbequina’. The total branching density (obtained by summing the branching density as in Figure 5 and the branching density of second-order, expressed as the number of lateral shoots more than 5 cm long inserted along the main branches, per cm of the central leader), was high in ‘I-77 self-pollinated’ (B), and ‘Fs-17 × Cipro’ (E) and with values statistically similar to that of ‘Arbequina’ (Figure 6).

The ratio between the diameter of the main branch and the diameter of the central leader (diameter ratio) was equal to 0.48 in ‘Arbequina’ (the lowest of all), while the highest value (0.64) was found in ‘Bouteillan × Nostrale di Rigali’ (D) (Figure 7). ‘Nucalia 1 × Don Carlo’ (L), ‘Nociara free pollinated 2’ (I), and ‘Fs 17 × Bouteillan’ (A) had values similar to that of ‘Arbequina’.

The ratio between the length of the main branch and its basal diameter ranged from 92 for ‘Arbequina’, to 66 for ‘Nociara free pollinated 2’ (I), about 29% less than the Spanish cultivar (Figure 8). Five genotypes (A, B, C, E, and L) had values similar to ‘Arbequina’. 

The diameter ratio was negatively correlated with branching frequency (Figure 9). 

## 3. Discussion

The aim of the present study was to evaluate the architectural characteristics of the canopy of the new olive genotypes, previously selected for high oil quality, in order to determine their suitability for use in new SHD plantations. The cultivar ‘Arbequina’ was used as control, as previous studies showed that it has tree architectural characteristics very suitable for SHD orchards [13,14,19,20,21]. Indeed, ‘Arbequina’ has a high branching frequency. This character is negatively correlated to the diameter of lateral branches as well as to the diameter of one-year-old shoots, enabling the branch/shoot biomass to be concentrated in a small volume (compact canopy). These are desirable characters for trees to be used in SHD olive orchards [13]. 

The average branch length along the central leader is an important parameter when planting olive trees at very high densities [22]. In fact, long branches create problems in olive orchard systems with short distances between rows because the branches occupy too much inter-row space. Furthermore, they can interfere with straddle machine harvesting, as the long lateral branches, which exceed the limit compatible with the harvest tunnel width, can be broken or damage the machine [21,22,23]. Therefore, for SHD olive orchards, it is preferable to have genotypes that develop lateral branches that are not excessively long. The average length of branches of the Arbequina cultivar was close to the average value across all genotypes. The genotype ‘Fs-17 × Bouteillan’ (A) had an average branch length much longer than all the other genotypes, on average 20 cm longer than those of ‘Arbequina’, while the genotype ‘I-79 free pollinated’ (G) had the shortest value, on average 9 cm less than Arbequina (Figure 1).

Rosati et al. [14] found that the branch insertion angle on the stem is not a relevant parameter in the characterization of 21 cultivars grown in a SHD olive orchard, but it gives an idea of the plant canopy habit. In the present study, ‘I-79 free pollinated’ (G), ‘Nociara free pollinated 1’ (H), and ‘Nociara free pollinated 2’ (I) had branch insertion angles around 50 degrees, which indicate an upright canopy habit, while all the other genotypes had higher angles, indicating an expanded canopy habit (Figure 3). In SHD olive orchards, narrow branch insertion angles could reduce light penetration into the internal parts of the canopy, with consequent reduction of the production efficiency and air circulation [23]. The branch insertion angle, in addition to being genotype dependent, is also affected by the space between the trees, in turn affected by the plantation distances, with wider angles found in wider distance plantations [24]. 

The branching frequency and the branching density are the most relevant parameters to assess whether a genotype is suitable for planting density intensification [13]. High values for these parameters imply a high ability to fill the available canopy volume with potentially productive shoots. In this trial, the genotype ‘Nucalia 1 × Don Carlo’ (L) had a branching frequency slightly higher, but not statistically different, than that of ‘Arbequina’ (Figure 4). The order of the genotypes according to the assumed value varied when branching density was evaluated (Figure 5). This is because the branching density is strictly dependent on the internode length. For instance, ‘Nucalia 1 × Don Carlo’ (L) no longer had the highest value, but rather ‘I-77 self-pollinated’ (B); this was because branching density indicates the number of branches per cm of the central axis length. The different internode lengths, which for genotype ‘I-77 self-pollinated’ (B) was about 1.6 cm, for ‘Nucalia 1 × Don Carlo’ (L) was about 2.0 cm, and for ‘Arbequina’ about 1.8 cm, help us to better understand these variations (Figure 2). The total branching density is one way to evaluate the canopy density, because it considers not only the main branches along the central leader but also the second-order lateral shoots present along the main branches (Figure 6). This is an interesting parameter to consider in establishing whether a genotype is suitable for super-high-density olive orchards. In fact, high branching values indicate that a given genotype can fill the available volume with numerous shoots, and therefore with potential fruiting sites [13,25,26]. In this study, only two genotypes, ‘Nociara free pollinated 1’ (H) and ‘Bouteillan × Nostrale di Rigali’ (D), had values statistically different from ‘Arbequina’ (Figure 6).

Another interesting parameter is the diameter ratio that denotes, with respect to the stem diameter, the thickness of the branch [13]. High diameter ratio values indicate that the branch has a relatively large diameter and therefore presumably is also more solid and above all rigid. This could be a drawback for continuous harvesting with straddle machines [21]. Thicker and stiffer branches can be broken more easily than thinner and more flexible ones. The diameter ratio also provides relevant information on how much a genotype invests in structural wood (stem, branches), i.e., in woody structures for supporting the potentially productive branches [14]. Having a low value of this parameter means that the plant allocates fewer resources to unproductive structures, thus making them available for other sites, such as reproductive ones [14,20]. ‘Arbequina’ was the only one to have a value less than 0.5 (Figure 7). However, other genotypes, namely ‘Nucalia 1 × Don Carlo’ (L), ‘Nociara free pollinated 2’ (I), and ‘Fs 17 × Bouteillan’ (A), had values statistically similar to ‘Arbequina’.

The ratio between branch length and its diameter can be considered an index of susceptibility of the branch being damaged by the harvester machine [21] (Figure 8). A high ratio indicates that a given genotype is potentially more suitable to resist the stresses generated during harvesting operations. Lodolini et al. [21] reported a value for ‘Arbequina’ (about 82 ± 12) that was on average lower than ours (about 92). Five genotypes, namely ‘Fs-17 × Bouteillan’ (A), ‘I-77 self-pollinated’ (B), ‘Nucalia 1 × Nostrale di Rigali’ (C), ‘Fs-17 × Cipro’ (E), and ‘Nucalia 1 × Don Carlo’ (L), had values similar to ‘Arbequina’.

The correlation between diameter ratio and branching frequency (Figure 9) is consistent with the results of Rosati et al. [13]. Moreover, the R^2^ of the present study showed even better values. The correlation confirms that genotypes that form multiple main branches along the central leader have branches that are relatively thinner. Therefore, genotypes characterized by high branching frequency have thinner branches, while genotypes with low branching have thicker branches. Hence, the correlation between diameter ratio and branching frequency is very important for potential adaptability of new genotypes to systems of plantation in SHD. Indeed, it is interesting to consider these aspects, because if a plant branches extensively, it means it fills the available volume with a greater number of potential productive sites. Additionally, thinner branches are more elastic and less prone to breakage by machinery during the olive harvesting phase. Arbequina showed the best correlation for the considered parameters together ‘Nociara free pollinated 2’ (I) and ‘Nucalia 1 × Don Carlo 1’ (L).

## 4. Materials and Methods

The genotypes used in this investigation were obtained in a breeding complex program, started around thirty-five years ago, which led to the establishment of the olive breeds collection plantation in “Tuoro sul Trasimeno” (Italy) in 2006 (https://biomemory.cnr.it), which comprises hundreds of breeds obtained from free and controlled pollination. In recent years, the biochemical profile of the olive oils obtained from these genotypes was carried out. Some results confirmed the possibility of improving and diversifying the quality of olive oil by crossbreeding known cultivars [9]. The best genotypes showing good biochemical composition of their VOOs in terms of beneficial compounds, such as fatty acid methyl esters (FAMEs) and phenolic substances, were then propagated to evaluate their agronomical characteristics and have been the object of the present study.

The trial was carried out in an olive orchard located in the Umbria region, in central Italy (Lat. 43°12′50″ N, Long. 12°03′20″ E, Alt. 287 m a.s.l.). In this geographical area, the annual average precipitation is around 870 mm, and annual average temperature is about 12.9 ± 5.7 °C [27].

Two-year-old olive trees, planted in 2020, were spaced 4 m × 2 m, with NE-SW row orientation. The plants were arranged in the field in 8 groups of 12 and 2 groups of 7 plants (genotypes E and D), each group containing plants of the same genotype. The trees were trained to a central leader and were not pruned, except for removal of basal shoots below 20 cm from the ground.

Soil management was carried out with tillage performed manually to avoid damage to the roots, starting in April and then once a month until September. Weed control is crucial to prevent weeds from competing with olive trees for nutrients and water. Plants were irrigated twice a week with a drip irrigation system from May to September. Fertilization was performed in April and May by using chemical fertilizers containing nitrogen, phosphorous, and potassium. The plantation was monitored for diseases and pests through frequent visual inspection of the vegetation and phytosanitary treatments against pests and diseases were performed when necessary.

Ten genotypes were selected and studied as well as peer-aged trees of ‘Arbequina’, a cultivar suitable for SHD orchards, which was used as control. For the purposes of the present work, each genotype was identified with a letter, as reported in Table 1. The characteristics of cultivars from which the breeds are derived are described in Olea databases [26], whereas ‘Nucalia 1’ and ‘Cipro’ are accessions of the “Consiglio Nazionale delle Ricerche, Istituto per i Sistemi Agricoli e Forestali del Mediterraneo”, Perugia, Italy.

The plants were obtained by rooting of cuttings taken from the mother plants cultivated in the main breed collection field. Each genotype consisted of twelve trees, except for two controlled crosses, ‘Bouteillan × Nostrale di Rigali’ (D) and ‘Fs-17 × Cipro’ (E), for which it was possible to plant only seven self-rooted plants due to the lower rooting aptitude of these two genotypes.

At the beginning of the trial, the average field value for the tree height was about 1.5 m and the number of main branches was 24. Measurements of architectural parameters were carried out on three representative trees per genotype, on the ‘central leader’, which is the main axis of the tree, the ‘main branches’, which are the ramifications inserted on the central leader, and the ‘shoots’, which are those inserted on the main branches. In particular, this study considered earlier research that suggests that small diameters and high branching are essential architectural factors to have compact canopies and high yield efficiency, which make genotypes potentially suitable for SHD orchards [13,15,19]. On the whole length of the central leader (except for the basal portion below 20 cm from ground), the following parameters were measured: height of central leader, number of nodes along the central leader, number of main branches (≥10 cm in length), length of the same main branches, node number along the main branches, and insertion angle of the branches. The internode length was determined by dividing the main branch length by the number of nodes on the same branch. This was performed for all branches. The branch insertion angle was measured in relation to the central leader, thus placing the zero of the goniometer parallel to the central leader of the tree.

Lateral shoots (>5 cm in length) were also counted along each of these branches. Furthermore, the basal diameter of all branches inserted along the central leader was measured (at the first internode) as well as the diameter of the central leader just below the branch considered (at the first internode below the insertion). The ratio between these two diameters was calculated [13]. The ratio between the length of the main branch and the diameter at the point of insertion with the central leader was calculated. The number of main branches is expressed as the number per bud or per cm of the central leader, as a measure, respectively, of branching frequency and branching density.

All the measurements were carried out in September 2022, except for the insertion angle, which was recorded in February 2023.

Data are presented as means ± standard errors. Genotype effects were analyzed by one-way analysis of variance (ANOVA) performed after normality and homoskedasticity tests. Averages were compared using Fisher’s LSD test (probability level < 0.05).

## 5. Conclusions

The results indicated that several genotypes have interesting architectural characteristics, such as branching frequency and density and diameter ratio, which were similar to those of the cultivar ‘Arbequina’. However, only the genotype ‘Nucalia 1 × Don Carlo’ (L) had values of all the considered architectural parameters similar to the cultivar ‘Arbequina’, making this genotype the most interesting among those evaluated. Such findings indicate that this genotype possesses architectural features making it potentially suitable for SHD orchards.

Although these results are promising, the study should be continued to evaluate additional aspects, such as yield potential and resistance to biotic/abiotic adversities, to have a complete characterization of the genotypes and, in particular, of ‘Nucalia 1 × Don Carlo’ (L).

## Figures and Tables

**Figure 1 plants-13-01399-f001:**
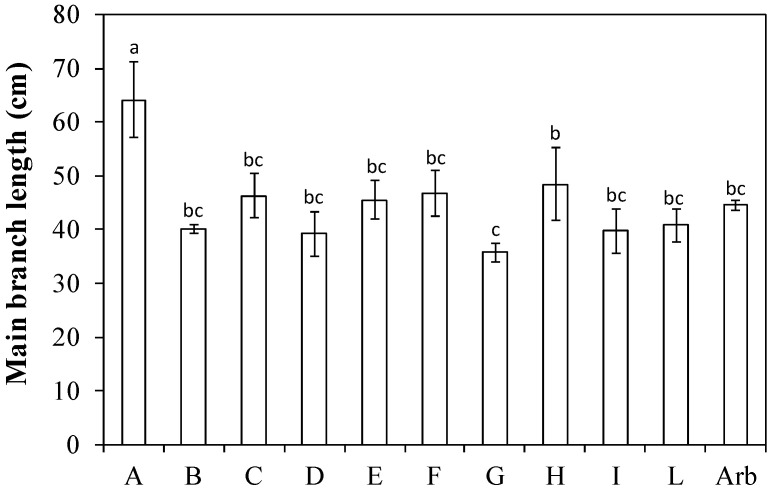
Length of the main branches in ten new olive genotypes and the ‘Arbequina’ cultivar. The letters on the *x* axis correspond to the breeds/genotypes listed in Table 1. Data represent averages and standard errors (bars). Different letters above the columns indicate significant differences between genotypes. Averages were compared using Fisher’s LSD test (*p* < 0.05).

**Figure 2 plants-13-01399-f002:**
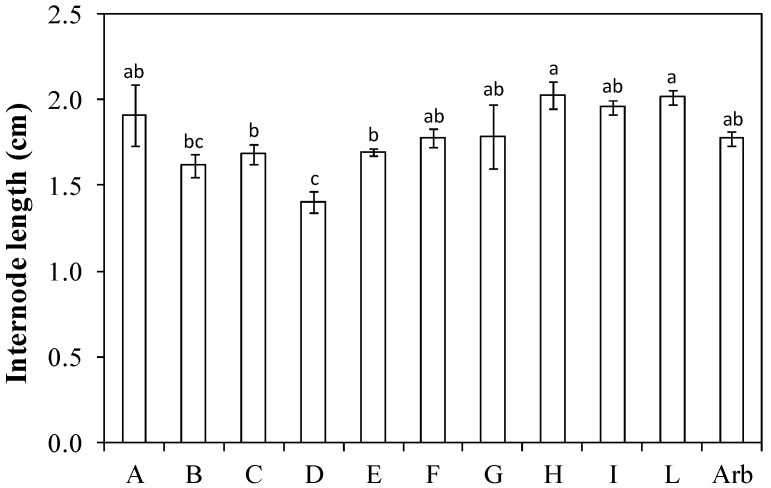
Internode length in ten new olive genotypes and the ‘Arbequina’ cultivar. The letters on the *x* axis correspond to the breeds/genotypes listed in Table 1. Data represent averages and standard errors (bars). Different letters above the columns indicate significant differences between genotypes. Averages were compared using Fisher’s LSD test (*p* < 0.05).

**Figure 3 plants-13-01399-f003:**
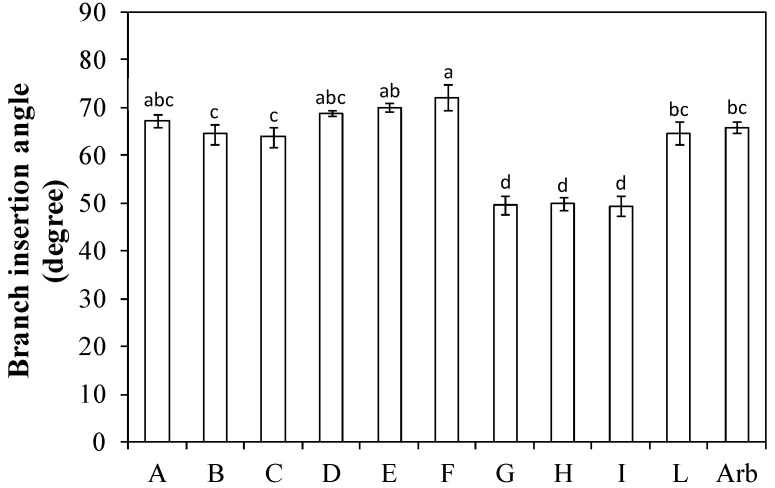
Branch insertion angle on the central leader in ten new olive genotypes and the ‘Arbequina’ cultivar. The letters on the *x* axis correspond to the breeds/genotypes listed in Table 1. Data represent averages and standard errors (bars). Different letters above the columns indicate significant differences between genotypes. Averages were compared using Fisher’s LSD test (*p* < 0.05).

**Figure 4 plants-13-01399-f004:**
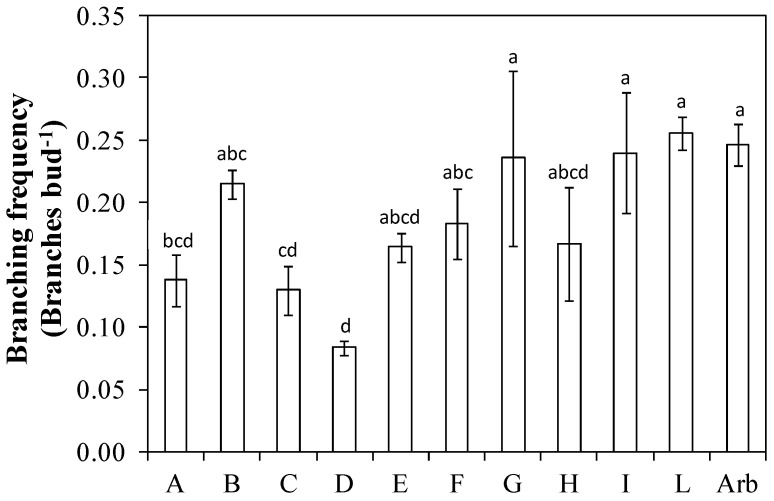
Branching frequency in ten new olive genotypes and the ‘Arbequina’ cultivar. The letters on the *x* axis correspond to the breeds/genotypes listed in Table 1. Data represent averages and standard errors (bars). Different letters above the columns indicate significant differences between genotypes. Averages were compared using Fisher’s LSD test (*p* < 0.05).

**Figure 5 plants-13-01399-f005:**
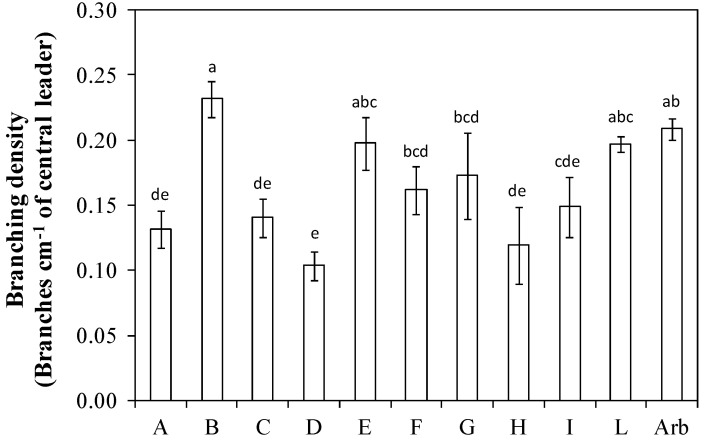
Branching density in ten new olive genotypes and the ‘Arbequina’ cultivar. The letters on the *x* axis correspond to the breeds/genotypes listed in Table 1. Data represent averages and standard errors (bars). Different letters above the columns indicate significant differences between genotypes. Averages were compared using Fisher’s LSD test (*p* < 0.05).

**Figure 6 plants-13-01399-f006:**
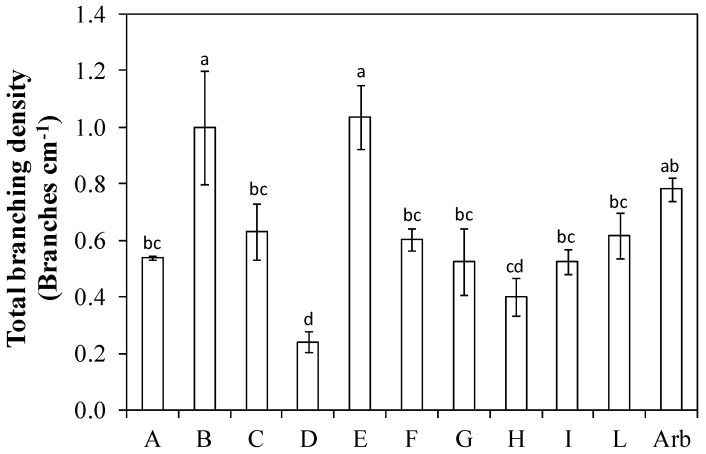
Total branching density in ten new olive genotypes and the ‘Arbequina’ cultivar. In the *y* axis, we considered the main branches and the shoots inserted on the main branches. The letters on the *x* axis correspond to the breeds/genotypes listed in Table 1. Data represent averages and standard errors (bars). Different letters above the columns indicate significant differences between genotypes. Averages were compared using Fisher’s LSD test (*p* < 0.05).

**Figure 7 plants-13-01399-f007:**
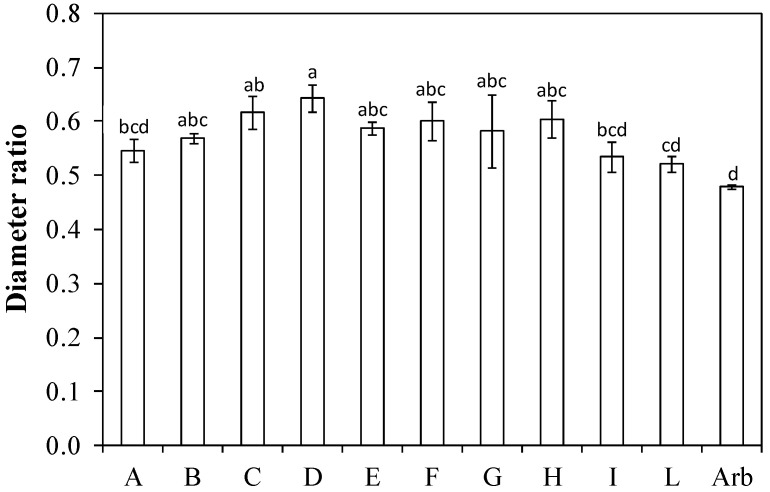
Diameter ratio (i.e., average ratio between main branch diameter and central leader diameter under the branch insertion) in ten new olive genotypes and the ‘Arbequina’ cultivar. The letters on the *x* axis correspond to the breeds/genotypes listed in Table 1. Data represent averages and standard errors (bars). Different letters above the columns indicate significant differences between genotypes. Averages were compared using Fisher’s LSD test (*p* < 0.05).

**Figure 8 plants-13-01399-f008:**
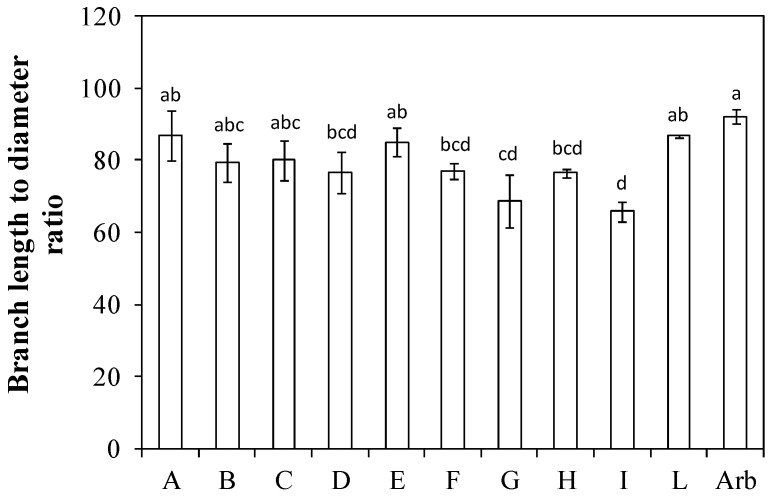
Branch length to diameter ratio (i.e., average ratio between main branch length and the diameter of the same branch measured at the insertion with the central leader) in ten new olive genotypes and the ‘Arbequina’ cultivar. The letters on the *x* axis correspond to the breeds/genotypes listed in Table 1. Data represent averages and standard errors (bars). Different letters above the columns indicate significant differences between genotypes. Averages were compared using Fisher’s LSD test (*p* < 0.05).

**Figure 9 plants-13-01399-f009:**
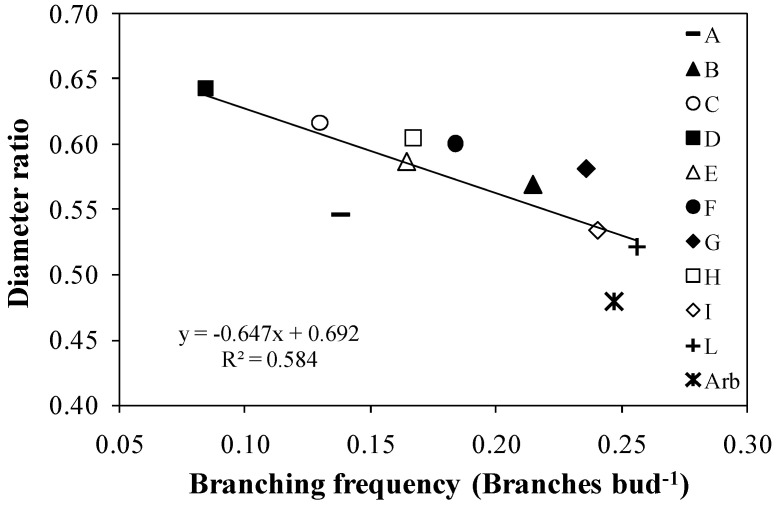
Correlation between the diameter ratio and the branching frequency in ten new olive genotypes and the ‘Arbequina’ cultivar. Each point is one of the genotypes listed in Table 1.

**Table 1 plants-13-01399-t001:** Identification letter of the genotypes under investigation.

Breed/Cultivar Name	Identification Letter	N° of Plants
Fs-17 × Bouteillan	A	12
I-77 self-pollinated	B	12
Nucalia 1 × Nostrale di Rigali	C	12
Bouteillan × Nostrale di Rigali	D	7
Fs-17 × Cipro	E	7
I-77 × Kalamata	F	12
I-79 free pollinated	G	12
Nociara free pollinated 1	H	12
Nociara free pollinated 2	I	12
Nucalia 1 × Don Carlo	L	12
Arbequina	Arb	12

## Data Availability

Data that are not already present on this manuscript will be made available upon reasonable request to the corresponding author (F.F.).

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
