# Peer review of "Canopy Architectural Characteristics of Ten New Olive (Olea europaea L.) Genotypes and Their Potential for Cultivation in Super-High-Density Orchards"

_plants, 2024, doi:10.3390/plants13101399_

Round 1

Reviewer 1 Report

Comments and Suggestions for Authors

The manuscript is focus in an interesting topic which is also very important from a practical point of view. Briefly, the authors compare simple architectural traits to compare the suitability of ten new genotypes of olive for super high density orchards (SHD). The study was well performed but the manuscript needs to be improved. Regarding statistical analysis the authors used an ANOVA and LSD as a post-test. LSD is too liberal when several treatments (in this case genotypes) are analyzed. Thus, considering that there are 10 genotypes compared against a control genotype using contrast or Dunnet could be more appropriate as differences between genotypes other than Arbequina are not considered in the manuscript.

Along the introduction and discussion most of the references are of articles from the same group with few citing of other groups works on olive and other fruit trees. I recommend to broaden the literature revision and include different points of views. The manuscript has many one sentence paragraph which is not appropriate for scientific writing. The English language and the tables and figures are adequate.

Introduction:  

The first four paragraphs of the introduction are not related to the topic of the manuscript. I suggest shortening that and include the description of the characters considered to be important for the adaptation to SHD and why are considered important.

Materials and Methods

Add more details. For example, were trees of the different genotypes randomly planted in the orchard or same genotypes were organize in plots?. In that case, how many trees per plot for each genotype and how many plots?. In case the trees of each genotype were mix within the orchard, how did you manage the situations in which one genotype was taller or had a greater growth rate and compete with those near by? Certain architectural traits (e.g. internode length, insertion angles..) can be modify in response to light.

Orientations of the rows

General dimension of the trees.

Give some information on the climate (mean temp, precipitation ETo), irrigation (related to the ETc)

For the measure variables, try to help the reader to make the connection between the traits of interests for suitability for SHD  mention in the introduction and what was measured

Conclusion

From the discussion I understood that no genotype was as good as Arbequina for SHD.  In the conclusion the message is that most evaluated genotypes were very good for SHD. This needs to be clarify

Below I am including a list of suggestions along the text

1)     Line 70 How variable was the average length along the tree?

2)     Line 278 Was this the average of all branches? Did you discriminate between heigh? and orientation? Would not be this information important for SHD adaptation?

3)     Line 73  In the M&M you mention that you measured the number of lateral branches. It is not presented in the results but it would be useful for the reader to have a general idea of how big the trees were and the number of branches would help.

4)     Figures: As Arbequina is your control genotype I suggest to change the M by Arb or something that would help to distinguish it from the other genotypes.

5)     Line 84 Internode length: was it measured? In which internodes? or was it estimated from the length and the number of nodes.? Add information in M&M

6)     Line 96. Again, how many branches per trees were measured? how variable this was within a tree? Did orientation or height influence this value?

7)     Line 109-111. This statement is not totally correct because only three genotypes were different from Arbequina (A, C, D), the rest all have an "a" indicating that were not different

8)     Line 122. Again. Revise the text according to the statistics. Also, improve the description presented in the text. Avoid paragraphs with only one sentence or two. Probably Branching density and frequency can be put together in a single paragraph.

9)     Lines 131-135 This explanation is not clear.  The branching density of the second-order, should not be over the cm of the branches instead of the central leader?

10)  Line 144. This is confusing. You talk about central leader, main axes and truck  for the same (?) thing in different parts of the text. Please unify criteria and explain in M&M what are you referring for each category of ramification

11)  Fig 9. I suggest to indicate (either with a different symbol or an arrow) which point correspond to Arbequina

12)  Line 180 Add references after “previous studies”

13)  Line 195. What does other authors outside the group say about branch insertion angle? What it is recommended in other fruit trees mechanically harvest?

14)  Line 209 That is not true. They have similar values

15)  Line 211 No so much like that. For example, D has short internodes, low branching frequency, and low branching density. Actually, all the paragraph needs to be to be improve including internode length in the discussion, and explaining the meaning of the different parameters.

16)  Line 223. Add the information of the genotypes in full name

17)  Line 255 Name of the branching are change again. Also, I understood you had divided the diameter of the branch by the diameter of the trunk. Here it is explained the other way round

18)  Line 245-248. This paragraph needs to be improve explaining first the meaning of the relation, then the comparison with Rosati, and later indication which genotypes were in the extreme of the relation and which looks similar to Arbequina

19)  Line 279 How did you measured the insertion angle? Is the angle measured in relation to the vertical, to the horizontal, or to the main trunk?

Author Response

We thank the reviewer for his/her suggestions and the time spent for evaluating/improving our paper.

We have addressed the reviewer’s requests.

Herewith, we list all the reviewer’s requests (in italics) and the changes made to the manuscript/answers given (in bold) to address the reviewer’s requests. In the revised version of the manuscript, all the corrections/changes made are highlighted in yellow.

Authors’ responses to comments of Reviewer 1

The manuscript is focus in an interesting topic which is also very important from a practical point of view. Briefly, the authors compare simple architectural traits to compare the suitability of ten new genotypes of olive for super high density orchards (SHD). The study was well performed but the manuscript needs to be improved. Regarding statistical analysis the authors used an ANOVA and LSD as a post-test. LSD is too liberal when several treatments (in this case genotypes) are analyzed. Thus, considering that there are 10 genotypes compared against a control genotype using contrast or Dunnet could be more appropriate as differences between genotypes other than Arbequina are not considered in the manuscript.

We used ANOVA and LSD as a posttest because, we think, they are acceptable for the scope of the study and we have familiarity with them. They have also been extensively used in previous similar researches. We understand that using contrast or Dunnet could be more appropriate as suggested by the Reviewer, but it would imply changes in the structure of the document and would create difficulties to us because we are not familiar with them. Therefore, we would prefer to keep the methods already used, also because we think that they allowed to reach the purpose of our research: to compare the different genotypes with the reference cultivar Arbequina. Also by the comments of Reviewer 2 the statistical analysis we performed seems acceptable. However, if Reviewer 1 think that the use of the suggested tests is essential we would try to find and involve a statistician as a co-author and try to use them.

Along the introduction and discussion most of the references are of articles from the same group with few citing of other groups works on olive and other fruit trees. I recommend to broaden the literature revision and include different points of views.

We have added four new articles. Please see articles 15-18 in the Reference list and lines 45-62 of the revised version of the paper.

The manuscript has many one sentence paragraph which is not appropriate for scientific writing. The English language and the tables and figures are adequate.

We have tried to improve this as much as possible along the whole article.

Introduction:  

The first four paragraphs of the introduction are not related to the topic of the manuscript. I suggest shortening that and include the description of the characters considered to be important for the adaptation to SHD and why are considered important.

As requested by the Reviewer, we have shortened the first four paragraphs of the Introduction and included the description of characters important for the adaptation to SHD systems. Please see lines 33-40 and 45-64 of the revised version of the paper. Moreover, we moved part of the text erased from the Introduction in Materials and Methods. Please see lines 262-272 of the revised version of the paper.

Materials and Methods

Add more details. For example, were trees of the different genotypes randomly planted in the orchard or same genotypes were organize in plots?

As requested by the Reviewer, we added more details. Please see lines 278-279 of the revised version of the paper.

In case the trees of each genotype were mix within the orchard, how did you manage the situations in which one genotype was taller or had a greater growth rate and compete with those near by? Certain architectural traits (e.g. internode length, insertion angles) can be modify in response to light.

As reported in the answer of the previous Reviewer’s comment, the trees were not mixed in the orchard and the measurements have been taken at an early stage for the plant when the distances between plants allowed each tree not to compete with/influence the neighboring ones.

Orientations of the rows

We added this information. See lines 277-278 of the revised version of the paper.

General dimension of the trees.

We added this information. Please see Lines 301-302 of the revised version of the paper.

Give some information on the climate (mean temp, precipitation ETo), irrigation (related to the ETc)

We added this information for the parameters that we found. Please see Lines 274-276 and reference n. 27 of the revised version of the paper.

For the measure variables, try to help the reader to make the connection between the traits of interests for suitability for SHD mention in the introduction and what was measured.

As requested by the Reviewer, we have inserted some text that make possible the connection between introduction and what measured reported in Materials and Methods. Please see/compare lines 45-64 and 301-313.

Conclusion

From the discussion I understood that no genotype was as good as Arbequina for SHD.  In the conclusion the message is that most evaluated genotypes were very good for SHD. This needs to be clarify

In the discussion we explained that diverse genotypes possess single characteristics comparable to Arbequina but only one genotype showed all the considered characteristics comparable to Arbequina. This is also reflected in the conclusions. However, we agree with the Reviewer that the conclusions were not clear. Now we better specified the results we obtained and hope that now the text is clearer. Please see lines 337-339 and 342-343 of the revised version of the paper.

Below I am including a list of suggestions along the text

1)     Line 70 How variable was the average length along the tree?

We measured the length of all branches (greater than or equal to 10 cm in length) along the central leader. We did not discriminate on the basis of the height where the branch was inserted along the central leader. We better specified this in Materials and Methods. Please see lines 308-313 of the revised version of the paper. The variability in the length of different branches can be estimated by the standard errors reported in Figure 1.

2)     Line 278 Was this the average of all branches? Did you discriminate between heigh? and orientation? Would not be this information important for SHD adaptation?

Yes, it was the average of all branches (see answer of previous Reviewer’s comment) inserted along the central leader, provided they were of length greater than or equal to 10 cm. We did not discriminate them on the basis of height of insertion or orientation. This is because in this preliminary phase of the work we focused on main general quantitative results. However, we think that the Reviewer’s suggestion is interesting and we will take it into account in the measurements that we will be performing in a second phase of our work aiming at a more detailed description of the considered genotypes.

3)     Line 73.  In the M&M you mention that you measured the number of lateral branches. It is not presented in the results but it would be useful for the reader to have a general idea of how big the trees were and the number of branches would help.

Please note that the number of lateral branches is not reported alone but in relation to both the number of buds (branching frequency) and the length of the central leader (branching density). Please see lines 321-323 of the revised version of the paper.

As far as the number of lateral shoots is concerned, as requested by the Reviewer, we reported this information. Please see lines 301-302 of the revised version of the paper.

4)     Figures: As Arbequina is your control genotype I suggest to change the M by Arb or something that would help to distinguish it from the other genotypes.

As suggested by the Reviewer, we replaced M with Arb throughout the revised version of the paper.

5)     Line 84 Internode length: was it measured? In which internodes? or was it estimated from the length and the number of nodes.? Add information in M&M

As requested by the Reviewer, we added this information. Please see lines 313-314 of the revised version of the paper.

6)     Line 96. Again, how many branches per trees were measured? how variable this was within a tree? Did orientation or height influence this value?

Please, see points 1, 2 and 3 above.

7)     Line 109-111. This statement is not totally correct because only three genotypes were different from Arbequina (A, C, D), the rest all have an "a" indicating that were not different

Our statement was limited to stating that those genotypes had a branching frequency value greater than 0.20. The significance is as the Reviewer has correctly reported.

We have added a sentence to better clarify the comparison among genotypes.  Please, see lines 112-115 of the revised version of the paper.

8)     Line 122. Again. Revise the text according to the statistics. Also, improve the description presented in the text. Avoid paragraphs with only one sentence or two. Probably Branching density and frequency can be put together in a single paragraph.

We have revised the text according to the Reviewer’s suggestion. Then, we have also grouped this paragraph with the following one. Please see lines 125-131 of the revised version of the paper.

9)     Lines 131-135 This explanation is not clear.  The branching density of the second-order, should not be over the cm of the branches instead of the central leader?

In Figure 6, we're not solely displaying the second-order branching density. Instead, we're presenting the total branching density, which encompasses both primary branching density (main branches directly stemming from the central leader) and second-order branching. Second-order branching includes all shoots measuring 5 cm or longer along the main branch. We applied this method to all main branches, ensuring a comprehensive representation of the entire tree, similar to what we did for first-order branching.

To maintain consistency, we expressed the shoots not per centimeter of the main branch's length but per centimeter of the central leader. This approach provides an index that relates to the same overall entity while simultaneously conveying the concept of canopy density.

10)  Line 144. This is confusing. You talk about central leader, main axes and truck  for the same (?) thing in different parts of the text. Please unify criteria and explain in M&M what are you referring for each category of ramification

As suggested by the Reviewer, we have uniformed the terms used. Please see lines 302-305 of the revised version of the paper.

11)  Fig 9. I suggest to indicate (either with a different symbol or an arrow) which point correspond to Arbequina

As suggested by the Reviewer, we highlighted each genotype with a unique symbol. Please see the new Figure 9 in the revised version of the paper.

12)  Line 180 Add references after “previous studies”

As suggested by the Reviewer, we added references. Please see line 181 of the revised version of the paper.

13)  Line 195. What does other authors outside the group say about branch insertion angle? What it is recommended in other fruit trees mechanically harvest?

There is not much literature on this topic. We found two articles: the number 22 and 23 in the reference list. The first states that wider insertion angles favor the penetration of air and light in the canopy improving the productive efficiency of the plants preventing at the same time diseases. The other one deals with the influence of the plantation distance on the angle of insertion. We took into consideration both. Please, see lines 200-205 of the revised version of the paper.

14)  Line 209 That is not true. They have similar values

As suggested by the Reviewer, we modified the sentence. Please see lines 209-211 of the revised version of the paper.

15)  Line 211 No so much like that. For example, D has short internodes, low branching frequency, and low branching density. Actually, all the paragraph needs to be to be improve including internode length in the discussion, and explaining the meaning of the different parameters.

We have tried to better explain the relationships between branching frequency and density. Please, see lines 212-213 of the revised version of the paper.

16)  Line 223. Add the information of the genotypes in full name

As requested by the Reviewer, we added the full names of genotypes. Please see lines 224-226 of the revised version of the paper.

17)  Line 255 Name of the branching are change again. Also, I understood you had divided the diameter of the branch by the diameter of the trunk. Here it is explained the other way round.

We changed the names as described in the previous point 10.

We could not find any of such info on line 255. Perhaps the Reviewer is referring to line 245 (= line 247 of the revised version of the paper). We confirm that we divided the diameter of the main branch by the diameter of the central leader. We have better discussed all this matter in the revised version of the paper and now, we hope, it should be clearer. Please see lines 249-259 of the revised version of the paper.

18)  Line 245-248. This paragraph needs to be improve explaining first the meaning of the relation, then the comparison with Rosati, and later indication which genotypes were in the extreme of the relation and which looks similar to Arbequina

As said in the previous point, we have improved this part according to the Reviewer’s suggestions. Please see lines 249-259 of the revised version of the paper.

19)  Line 279 How did you measured the insertion angle? Is the angle measured in relation to the vertical, to the horizontal, or to the main trunk?

We have specified this in Materials and Methods. Please see lines 325-327 of the revised version of the paper.

Reviewer 2 Report

Comments and Suggestions for Authors

Dear sirs,

thank you for submitting your research. I found it quite interesting and well conducted.
During my revision, i have not found significant issues in your research, except two minor but important points.

1- statistical data treatment in Figures from 1 to 8 is done by ANOVA, yet to carry out ANOVA analysis data MUST have a normal distribution, and I have not read anything about normality and homoskedasticity tests. 

2- Figure 9 reports a R2 = 0.5846 that is quite low and, in my opnion,  represent a weak correlation. I strongly suggest to find another method to search for correlation. Moreover, the spots in Fig. 9 cannnot identify the different cultivars and so the figure could lead to misunderstandings.

No further suggestions.

Author Response

We thank the reviewer for his/her suggestions and the time spent for evaluating/improving our paper.

We have addressed the reviewer’s requests.

Herewith, we list all the reviewer’s requests (in italics) and the changes made to the manuscript/answers given (in bold) to address the reviewer’s requests. In the revised version of the manuscript, all the corrections/changes made are highlighted in yellow.

Authors’ responses to comments of Reviewer 2

Dear sirs,

thank you for submitting your research. I found it quite interesting and well conducted.

We thank the Reviewer for his/her positive comments on our paper.

During my revision, i have not found significant issues in your research, except two minor but important points.

  • statistical data treatment in Figures from 1 to 8 is done by ANOVA, yet to carry out ANOVA analysis data MUST have a normal distribution, and I have not read anything about normality and homoskedasticity tests. 

The software used for the statistical analyses (ANOVA analysis) automatically performs a preliminary normality and homoskedasticity test. We added a few lines in Materials and Methods to say this. Please see lines: 329-330 of the revised version of the paper.

  • Figure 9 reports a R2 = 0.5846 that is quite low and, in my opnion, represent a weak correlation. I strongly suggest to find another method to search for correlation. Moreover, the spots in Fig. 9 cannnot identify the different cultivars and so the figure could lead to misunderstandings.

Even though the relationship is not extremely high, considering the variability of the parameters taken into account, we think that it can be considered good and in line with previous literature on the subject (Rosati et al. 2013). Then, according to the suggestion of the Reviewer we identified the different cultivars in the figure by using specific symbols and adding a legend. Please see the revised version of Figure 9.

Round 2

Reviewer 1 Report

Comments and Suggestions for Authors

I want to thank the authors for the taking in consideration most of my comments in the previous version. I think that the manuscript has improve sustantially. I have some minor suggestions to consider

1) in line 97 add "(i.e., more vertical)" and line line 100 add "(i.e., more horizontal)"

2) Fig 5 in the Y axis add " (Branches cm-1 of central leader trunk)

3) Fig 6 in the Y axis you have "Total branching density (branches/Shoot cm-1)". This sounds strange because "shoots" are ramification of the higher order (i.e., newer) than the branches.  Please revise  

4) Discussion. I recommend to put together paragraph one and two. Also, add a mention to branch lenght in Arbequina. Otherwise you are explaining the characters that make Arbequina suitable for SHD and then the first character you describe for the other genotypes have not been mention in the previous paragraph which is not helping the reader

Author Response

We thank again the reviewer for his/her suggestions and the time spent for evaluating/improving our paper.

We have addressed the reviewer’s requests.

Herewith, we list all the reviewer’s requests (in italics) and the changes made to the manuscript/answers given (in bold) to address the reviewer’s requests. In the revised version of the manuscript, all the corrections/changes made are highlighted in yellow.

Authors’ responses to Reviewer’s comments

1) in line 97 add "(i.e., more vertical)" and line line 100 add "(i.e., more horizontal)"

Done. Please see lines 96-97 and 99 of the revised version of the paper.

2) Fig 5 in the Y axis add " (Branches cm-1 of central leader trunk)

Done. Please see the new Figure 5 in the revised version of the paper. We added in the Y axis “Branches cm-1 of central leader”. We did not mention “trunk” since we always used the term “central leader” in the article and not central leader trunk.

3) Fig 6 in the Y axis you have "Total branching density (branches/Shoot cm-1)". This sounds strange because "shoots" are ramification of the higher order (i.e., newer) than the branches.  Please revise  

Done. Please see the new version of Figure 6 in the revised version of the paper. We added an explanation (please see lines 139-140): “In the Y axis we considered the main branches and the shoots inserted on the main branches”.

4) Discussion. I recommend to put together paragraph one and two. Also, add a mention to branch lenght in Arbequina. Otherwise you are explaining the characters that make Arbequina suitable for SHD and then the first character you describe for the other genotypes have not been mention in the previous paragraph which is not helping the reader.

Done. Please see line 179 and lines: 186-197 of the revised version of the paper.

Other changes made on the text:

Line 125: we added “‘I-77 × Kalamata’ (F), and ‘I-79 free pollinated’ (G)”

We took the opportunity to make some minor adjustments on the text correcting mainly punctuation and words wrongly written. We highlighted in yellow the adjustments that we made.